# Transcriptome and Metabolome Analyses Reveal Complex Molecular Mechanisms Involved in the Salt Tolerance of Rice Induced by Exogenous Allantoin

**DOI:** 10.3390/antiox11102045

**Published:** 2022-10-18

**Authors:** Juan Wang, Yingbo Li, Yinxiao Wang, Fengping Du, Yue Zhang, Ming Yin, Xiuqin Zhao, Jianlong Xu, Yongqing Yang, Wensheng Wang, Binying Fu

**Affiliations:** 1Institute of Crop Sciences, Chinese Academy of Agricultural Sciences, Beijing 100081, China; 2College of Life Sciences, China Agricultural University, Beijing 100193, China; 3College of Agronomy, Anhui Agricultural University, Hefei 230036, China; 4National Nanfan Research Institute (Sanya), Chinese Academy of Agricultural Sciences, Sanya 572024, China

**Keywords:** allantoin, rice, salt stress, antioxidant, transcriptome, metabolite

## Abstract

Allantoin is crucial for plant growth and development as well as adaptations to abiotic stresses, but the underlying molecular mechanisms remain unclear. In this study, we comprehensively analyzed the physiological indices, transcriptomes, and metabolomes of rice seedlings following salt, allantoin, and salt + allantoin treatments. The results revealed that exogenous allantoin positively affects the salt tolerance by increasing the contents of endogenous allantoin with antioxidant activities, increasing the reactive oxygen species (ROS)–scavenging capacity, and maintaining sodium and potassium homeostasis. The transcriptome analysis detected the upregulated expression genes involved in ion transport and redox regulation as well as the downregulated expression of many salt-induced genes related to transcription and post-transcriptional regulation, carbohydrate metabolism, chromosome remodeling, and cell wall organization after the exogenous allantoin treatment of salt-stressed rice seedlings. Thus, allantoin may mitigate the adverse effects of salt stress on plant growth and development. Furthermore, a global metabolite analysis detected the accumulation of metabolites with antioxidant activities and intermediate products of the allantoin biosynthetic pathway in response to exogenous allantoin, implying allantoin enhances rice salt tolerance by inducing ROS scavenging cascades. These results have clarified the transcript-level and metabolic processes underlying the allantoin-mediated salt tolerance of rice.

## 1. Introduction

Allantoin, a key intermediate of purine catabolism, is ubiquitous among plants and is important for plant growth and development. Allantoin and its hydrolyzed product allantoate are ureides, which are crucial nitrogenous compounds that are translocated in plants from the source organs to the sink organs [1]. Additionally, allantoin is the main source of nitrogen, and its content is spatiotemporally regulated, with substantial increases during the reproductive stage in nitrogen-fixing legumes and nitrate-feeding legumes [2]. Moreover, it is the sole source of nitrogen in non-leguminous plants under nitrogen-deficient conditions [3,4].

In addition to its role in nitrogen metabolism, allantoin also substantially contributes to abiotic stress tolerance. Earlier research revealed that the allantoin content varies among rice genotypes, and the allantoin level in grains is positively correlated with the tolerance of rice seedlings to chilling and water deficit stresses [5]. Furthermore, allantoin is a positive metabolic marker of salt tolerance in rice [6]. More specifically, salt-tolerant rice genotypes accumulate more allantoin in the roots than salt-sensitive genotypes [7]. Allantoin accumulation in response to abiotic stresses has been observed in Arabidopsis [8,9], wheat [10], barley [11], and legumes [12].

Increases in the allantoin content in various plants exposed to abiotic stress suggest allantoin positively regulates stress tolerance. A few studies demonstrated that allantoin can function as an antioxidant that mitigates oxidative damage in plants under stress conditions [13,14]. The accumulation of allantoin in the Arabidopsis *aln* (allantoinase) mutant, which is due to decreases in *ALN* expression and ALN enzymatic activity, coincides with the activation of antioxidant pathways in response to Cd treatments [8,15]. The in vitro antioxidant activity of plant extracts containing allantoin proved that allantoin had antioxidant properties and had a positive effect on total antioxidant capacity in plants [16]. However, in rice, allantoin functions more as a signaling molecule rather than an antioxidant that scavenges free radicals [5]. Whether allantoin is an antioxidant that mediates rice stress tolerance remains to be conclusively determined.

The molecular mechanisms underlying allantoin-mediated abiotic stress tolerance have been investigated. Watanabe et al. analyzed the genome-wide gene expression of *aln* mutants under drought and osmotic stress conditions, and the generated data indicated that decreases in allantoinase gene expression levels induce the accumulation of allantoin and increase the expression of stress-related genes, especially abscisic acid (ABA) biosynthesis genes, suggesting ABA is involved in allantoin-mediated abiotic stress responses in Arabidopsis [17]. In another study, jasmonic acid (JA) signaling pathway genes were activated in the Arabidopsis *aln* mutant, and further analyses revealed that the allantoin-enhanced JA response was dependent on ABA [18]. Considered together, these findings imply that allantoin may coordinately regulate ABA and JA pathways in response to abiotic stresses.

The utility of several biochemical-based strategies for enhancing plant abiotic stress tolerance has been assessed via the exogenous application of stress signaling molecules or natural compounds [19]. In the present study, exogenous allantoin was applied to rice to elucidate its physiological roles and regulatory effects on molecular mechanisms mediating salt tolerance. Furthermore, a multi-omics analysis revealed transcriptome and metabolome changes in rice in response to exogenous allantoin under salt stress conditions. The results of this study provide important insights into the molecular basis of rice salt tolerance.

## 2. Materials and Methods

### 2.1. Rice Materials and Stress Treatments

A moderate salt-tolerant rice variety, C18 (*Oryza sativa* L. ssp. *indica*), was used in this study. Seeds of C18 were surface-sterilized in 1% NaClO (*v*/*v*) for 10 min. They were subsequently washed, soaked in water at 25 °C for 24 h, and germinated in an incubator set at 37 °C for 24 h. The germinated seeds were placed in 96-well PCR plates with the bottom clipped and then cultured in water for 7 days. The seedlings were transferred to Yoshida nutrient solution and grown in a phytotron (size: 4.0 m × 2.5 m × 2.5 m) with a day (30 °C, 14 h)/night (22 °C, 10 h) cycle, an irradiance of about 700 mmol quanta m^−2^s^−1^, and a minimum relative humidity of 70%. When they reached the three-leaf stage, the seedlings underwent the following treatments: control (Yoshida nutrient solution), high salinity (Yoshida nutrient solution supplemented with 140 mM NaCl), allantoin (Yoshida nutrient solution supplemented with 10 mM allantoin), and salt + allantoin (Yoshida nutrient solution supplemented with 140 mM NaCl and 10 mM allantoin). The salt concentration was selected based on our preliminary tests where 140 mM NaCl application led to significant toxicity symptoms including leaf chlorosis and rolling; 144 rice seedling plants were used for each treatment, each treatment was replicated three times. Shoot and root samples were collected 24 h after initiating the treatments for the transcriptome analyses, whereas samples were collected after 3 days for the analyses of the metabolome and physiological indices. All collected samples were frozen immediately in liquid nitrogen and stored at −80 °C.

### 2.2. Measurement of Physiological Indices

After the plants were treated for 14 days, they were transferred to Yoshida nutrient solution for a 7-day recovery period. The seedling survival rates were calculated as the surviving-plants-to-total-number-of-treated-plants ratio. The sodium and potassium concentrations were determined as previously described [20]. The proline, malondialdehyde (MDA), peroxidase (POD), and superoxide dismutase (SOD) concentrations were measured using physiological assay kits (Suzhou Grace Biotechnology Co., Ltd., Suzhou China). Each experiment was independently repeated three times. Shoots and roots of rice seedlings were homogenized on ice with a mortar and pestle in extracting solution. The homogenate was centrifuged at 12,000× *g* for 10 min at 4 °C. The supernatant was used immediately for proline, malondialdehyde content, and enzyme assays. For the proline content, plant samples were extracted with sulfosalicylic acid and then heated with acid ninhydrin to form a red substance. The concentration of proline was determined by measuring the absorbance at 520 nm. MDA was determined with the thiobarbituric acid method, at high temperature and acid conditions, and the MDA and thiobarbituric acid shrank, resulting in red products with maximum absorption at 532 nm. The MDA content of the supernatant was determined at 532 and 600 nm by spectrophotometer. The activity of SOD was measured according to a method using xanthine oxidase and WST-8. WST-8 could react with O^2−^ catalyzed by Xanthine Oxidase (XO) to produce soluble formazan dyes with maximum absorption at 450 nm. SOD can remove O^2−^ and inhibit the formation of formazan. One unit of SOD was defined as the amount of enzyme that inhibits the rate of formazan dyes reduction by 50%. The activity of POD was determined using the guaiacol oxidation method. Under the catalysis of peroxidase, H_2_O_2_ oxidizes guaiacol to generate reddish-brown product, which has the maximum absorption at 470 nm. One unit of POD activity was defined as the amount of enzyme required for each 1 increase in absorbance at 470 nm in 1 min.

### 2.3. Quantitative Analysis of Allantoin

Leaf samples were ground in liquid nitrogen and then resuspended in 1.5 mL ddH_2_O. Each sample was sonicated at 60 °C for 40 min and then centrifuged. The supernatant was collected and filtered using the Waters OASIS HLB 3CC 60 MG 100 BX extraction column. The allantoin content was determined using a high-performance liquid chromatography system. The analytes were separated using the Shimadzu NH2 column (250 mm × 4.6 mm, 5 μm), with water (solvent A) and methanol (solvent B) (90:10) as the mobile phases. The flow rate was 1.0 mL/min and the column temperature was 30 °C. A 10 μL aliquot of each sample was injected for the analysis.

### 2.4. Examination of the Allantoin Antioxidant Activity

The in vitro concentration-dependent antioxidant activity of allantoin was evaluated using the DPPH Free Radical-Scavenging Capacity Assay kit (Solarbio Technology Co., Ltd., Beijing, China), the ABTS Free Radical-Scavenging Capacity Assay kit (Solarbio Technology Co., Ltd.), and the Total Antioxidant Capacity (T-AOC) Assay kit (Solarbio Technology Co., Ltd.). Allantoin was used at doses of 0.01, 0.025, 0.05, 0.1, and 0.15 mg/mL in the assays. An authentic antioxidant (vitamin C) used at the same doses as allantoin served as the control to confirm the utility of the assays. All measurements were completed using three independent biological replicates.

### 2.5. Transcriptome Sequencing and Data Analysis

Total RNA was extracted using the TRIzol reagent. The concentration and purity of the extracted RNA were determined using the NanoDrop 2000 spectrophotometer (Thermo Fisher Scientific, Wilmington, DE, USA). First-strand cDNA was synthesized using a random hexamer primer and M-MuLV Reverse Transcriptase. The constructed libraries were sequenced on an Illumina platform, which generated paired-end reads. The raw reads were processed using the BMKCloud (www.biocloud.net) (accessed on 15 June 2022) online bioinformatics platform. The adapter sequences and low-quality reads were removed from the data sets. The retained clean reads were mapped to the reference genome sequence using the HISAT2 software (Center for Computational Biology, Johns Hopkins University, Baltimore, MD, USA). To ensure the expression levels of genes at different loci on different chromosomes could be compared, FPKM values were calculated. The differentially expressed genes (DEGs) were identified using nbinomTest of the DESeq R package and the following criteria: fold-change ≥ 1.5 and *p* < 0.05. Gene functions were determined according to the KEGG Ortholog database. A GO enrichment analysis was performed using the web-based tool agriGO (v2.0) (http://systemsbiology.cau.edu.cn/agriGOv2/index.php) (accessed on 21 June 2022).

### 2.6. Metabolome Analysis and Data Analysis

Biological samples were lyophilized using the Scientz-100F freeze-dryer. The lyophilized material (100 mg) was resuspended with 1.2 mL 70% methanol solution, vortexed for 30 s every 30 min (six times in total), and centrifuged at 12,000 rpm for 10 min. The supernatant was filtered (SCAA-104, 0.22 μm pore size; ANPEL, Shanghai, China) and then analyzed using a UPLC-ESI-MS/MS system (UPLC: Nexera X2, Shimadzu, Kyoto, Japan; MS: Applied Biosystems 4500 QTRAP, Applied Biosystems, Waltham, MA, USA). The GC-MS data sets were imported separately into the SIMCA-P^+^ 14.0 software package (Umetrics, Umeå, Sweden). Significant metabolites (VIP > 1.0 and *p* < 0.05) were annotated on the basis of available reference standards, the NIST 11 standard mass spectral databases, and the Fiehn database linked to the ChromaTOF software (ChromaTOF Version 5.0, LECO, St. Joseph, MO, USA). This analysis was completed using three independent biological replicates. Student’s *t*-test (*p* < 0.05) was used to identify significant differentially abundant metabolites (DAMs).

### 2.7. Quantitative Real-Time (qRT)-PCR Analysis

Total RNA was extracted using the TRNzol Universal Reagent (Tiangen Biochemical Technology Co., Ltd., Beijing, China). The RNA was used as the template for synthesizing cDNA using the Reverse Transcription Kit for the Synthesis of First-strand cDNA (Tiangen Biochemical Technology Co., Ltd.). The qRT-PCR primer sequences were designed using the Primer 5 software (PREMIER Biosoft International, San Francisco, USA) and checked by conducting a BLAST search of the rice genome database. Relative expression levels were calculated according to the 2^−∆∆Ct^ method. Each experiment was independently repeated three times. Details regarding the analyzed genes and the gene-specific primers are listed in Appendix A.

### 2.8. Analysis of the Correlation between the Transcriptome and Metabolome Data

The correlation between the metabolome and transcriptome data was analyzed to clarify the mechanism underlying the transcriptional regulation of metabolic pathways. On the basis of the results of the analyses of differences in metabolite abundances and gene expression levels, the DEGs and DAMs in the same group were simultaneously mapped to KEGG pathways. The significant pathways (*p* < 0.05) were selected for further analyses. The correlation between gene expression and metabolite abundance was analyzed according to the Pearson correlation method. The data were preprocessed via the Z-value transformation method. The following criteria were used to identify significant correlations: correlation coefficient > 0.80 and *p* < 0.05. The DAMs and DEGs were classified in each group using the K-means algorithm and then mapped according to the classification. Consistent trends in the changes to metabolite abundances and gene expression levels were detected, which may reflect certain relationships between these metabolites and genes.

## 3. Results

### 3.1. Exogenous Allantoin Improves the Salt Tolerance of Rice Seedlings

To investigate the physiological effects of exogenous allantoin on rice plants exposed to salt stress, C18 seedlings at the three-leaf stage were treated with 140 mM NaCl, 10 mM allantoin, or 140 mM NaCl + 10 mM allantoin for 7 days. Unlike the control leaves, the leaves of the C18 seedlings turned yellow and were curled under salt stress conditions (Figure 1A). In contrast, there were no phenotypic differences between the control and allantoin-treated plants. Moreover, the salt stress-induced symptoms were substantially alleviated by the salt + allantoin treatment. The survival rate was significantly higher for the seedlings that underwent the salt + allantoin treatment (79.16%) than for the seedlings treated with salt alone (47.22%) (Figure 1B).

Stress-related parameters, including proline and MDA contents and SOD and POD antioxidant activities, were investigated under the four treatment conditions to clarify whether allantoin increases rice seedling salt tolerance. The proline and MDA contents increased in the shoots and roots under salt stress conditions (Figure 1C,D). However, the proline content in the shoots and roots were higher following the salt + allantoin treatment than after the salt treatment (Figure 1C). Conversely, the MDA content in the shoots was lower following the salt + allantoin treatment than after the salt treatment (Figure 1D). The activities of SOD and POD in the shoots were significantly inhibited by salt stress, but exogenous allantoin increased the activities of both enzymes. The SOD and POD activities in the shoots and roots were higher in the seedlings treated with salt + allantoin than in the seedlings treated with salt alone (Figure 1E,F). These results demonstrated that the application of exogenous allantoin alleviated the oxidative damage in rice seedlings exposed to salt stress.

We further examined the K^+^ and Na^+^ contents in the C18 seedlings under the four treatment conditions. Compared with the effects of the salt treatment, the salt + allantoin treatment resulted in an increase in the K^+^ and Na^+^ contents in the roots but a decrease in the Na^+^ content in the shoots (Figure 1G,H), reflecting the differential effects of exogenous allantoin on the shoots and roots under saline conditions. Thus, the Na^+^/K^+^ ratio decreased only in the roots in response to exogenous allantoin (Figure 1I). Considered together, these results indicate that exogenous allantoin may increase rice seedling salt tolerance by maintaining the Na^+^ and K^+^ homeostasis.

The examination of the endogenous allantoin content in C18 seedlings in response to different treatments revealed that salt stress and the application of exogenous allantoin increased the endogenous allantoin content in the shoots and roots, relative to the control level. Additionally, the exogenous allantoin treatment considerably increased the endogenous allantoin content in the roots of C18 seedlings under salt stress conditions (Figure 1J). These results suggest that allantoin positively affects rice responses to salt stress.

### 3.2. Antioxidant Activity of Allantoin In Vitro

The antioxidant activity of allantoin is reportedly similar to that of vitamin C [21]. To determine whether allantoin functions as an antioxidant in rice, we compared allantoin and ascorbic acid in terms of their ability to scavenge DPPH and ABTS free radicals and their T-AOC, which was determined on the basis of the reduction of Fe^3+^. The DPPH free radical, which is very stable, is widely used for studying antioxidants. We observed that both allantoin and ascorbic acid had antioxidant activities. However, at low concentrations, vitamin C was better able to scavenge DPPH and ABTS free radicals than allantoin. When the concentration increased from 0.01 to 0.05 mg/mL, the DPPH free radical scavenging rate of allantoin increased from 4.29% to 14.55% (Figure 2A), whereas the ABTS free radical scavenging rate increased from 0.53% to 3.81% (Figure 2B). The results of the Fe^3+^ reduction experiment indicated that the reducing activity of vitamin C increased as the concentration increased, but there was no change in the reducing activity of allantoin (Figure 2C). Hence, allantoin can scavenge free radicals, but its antioxidant activity is lower than that of vitamin C.

### 3.3. Transcriptome Profiling of Rice Seedlings following Different Stress Treatments

To identify genes involved in rice responses to salt stress and exogenous allantoin, a transcriptome sequencing (RNA-seq) analysis was performed to identify the DEGs among treatments. The comparison with the C18 seedlings under control conditions revealed 3627 (2219 upregulated and 1408 downregulated), 7406 (4560 upregulated and 2846 downregulated), and 289 (160 upregulated and 129 downregulated) DEGs in the seedlings after the salt, allantoin, and salt + allantoin treatments, respectively (Figure 3A). Therefore, the treatments with salt or allantoin alone altered gene expression in rice considerably more than the salt + allantoin treatment. A total of 24 DEGs were randomly selected to verify the transcriptome sequencing results. The qRT-PCR data for these 24 genes were consistent with the RNA-seq data (*r*^2^ = 0.896; Appendix A), indicative of the reliability of the transcriptome sequencing results.

We performed a GO enrichment analysis of the genes that were expressed at higher or lower levels in the C18 seedlings treated with allantoin than in the control seedlings (Figure 3A, Appendix A). The enriched GO terms among the 4560 upregulated genes were translation, carbohydrate metabolic process, gene expression, oxidation reduction, and signal transduction. The enriched GO terms assigned to the 2846 downregulated genes were protein modification process, transport, photosynthesis, and response to stimulus (Appendix A). Accordingly, the rice genes that were differentially regulated by exogenous allantoin had diverse functions.

### 3.4. Unique Sets of Genes Related to Allantoin-Mediated Salt Stress Tolerance

Compared with the salt-treated C18 seedlings, 114 and 656 genes had up- and downregulated expression levels, respectively, in the seedlings that underwent the salt + allantoin treatment (Figure 3A, Appendix A), reflecting the suppressive effects of exogenous allantoin on gene expression in rice seedlings exposed to salt stress. The significantly enriched GO terms among the upregulated genes were transmembrane transport and oxidoreductase activity (Figure 3B). The further functional characterization of the genes indicated the upregulated genes associated with transmembrane transport included *OsALMT4*, *OsFPPS1*, *OsCBSCLC2*, *OsBAT4*, *OsNAAT1*, *OsYSL2*, *OsMATE13*, *OsYSL16*, *OsHKT4*, *OsSPS*, *OsZIP4*, *OsZIP8*, *OsNRAMP1*, *OsGAT4*, and *OsNTP8*. Moreover, the 14 upregulated genes related to oxidoreductase activity included *OsALDH2C1*, *OsDWARF4*, *OsHSL6*, *OsTRXh1*, and *APX6* (Appendix A). These results suggest that the exogenous application of allantoin enhances transport functions and ROS homeostasis in salt-stressed rice seedlings. The enriched GO terms assigned to the downregulated genes in the C18 seedlings treated with both salt and allantoin were carbohydrate metabolic process, post-translational protein modification, protein kinase activity, protein-DNA complex assembly, and nucleotide binding (Figure 3C), suggesting that these biological processes are repressed by exogenous allantoin.

We analyzed the DEGs detected by the salt stress vs. control and salt + allantoin vs. salt stress comparisons by constructing a Venn diagram. Most of the genes (608 of 775) that were expressed at lower levels after the salt + allantoin treatment than after the salt treatment were expressed at higher levels after the salt treatment than after the control treatment (Figure 3D), indicating that exogenous allantoin mainly repressed the expression of the genes that were upregulated by salt stress. Notably, the downregulated genes included those encoding 46 protein kinases (7 *OsRLCK* genes, 10 *OsPK* genes, 19 *OsPTK* genes, *OsBAK1*, *OsSPARK2*, and several *OsRLK* genes), 33 glycosyl hydrolase family proteins (e.g., 4 *OsXTH* genes, 5 *OsBGAL* genes, 3 *OsBGLU* genes, and glycosyl hydrolase subfamily members), 33 transcription factors (e.g., 8 *OsbHLH* genes, 7 *OsMYB* genes, 3 *OsERF* genes, 4 *OsHOX* genes, *OsWRKY34*, *OsNAC6*, *OsbZIP46*, and *OsPCF5*), 10 histone proteins (3 *H2A* genes, 4 *H3* genes, 2 *H4* genes, and *H2B.2*), and 10 cellulose synthases and expansins (2 *OsCESA* genes, 2 *OsEXPA* genes, 4 *OsEXPB* genes, and 2 *OsCLSF* genes) (Appendix A).

### 3.5. Metabolome Changes in Rice Seedlings That Underwent the Salt or Salt + Allantoin Treatments

To investigate the effects of exogenous allantoin on the metabolites in rice seedlings in response to salt stress, leaves were collected from C18 seedlings following the control, allantoin, salt, and salt + allantoin treatments for a global metabolite profiling analysis. Pearson’s correlation analysis confirmed the reproducibility among the biological samples (Figure 4A). A principal component analysis was performed to elucidate the overall metabolic differences between each group and the degree of variation between the samples within each group. The four groups of samples were separated and clustered into distinct groups (Figure 4B). Accordingly, the reproducibility between our samples was sufficient for further analyses.

The comparison with the control C18 seedlings detected 192 DAMs (79 and 113 with increased and decreased abundances, respectively) in the allantoin-treated seedlings (Figure 4C, Appendix A), suggesting that the exogenous allantoin treatment adversely affected metabolite accumulation in rice seedlings. The enriched KEGG pathways among the DAMs were biosynthesis of amino acids, ABC transporters, aminoacyl-tRNA biosynthesis, 2-oxocarboxylic acid metabolism, and tryptophan metabolism (Appendix A).

The comparison with the salt-treated C18 seedlings revealed 60 DAMs (27 and 33 with increased and decreased abundances, respectively) in the seedlings that underwent the salt + allantoin treatment (Figure 4C,E, Appendix A). The enriched KEGG pathways among these DAMs were glutathione metabolism, carbon metabolism, aminoacyl-tRNA biosynthesis, and purine metabolism (Figure 4D). Of these DAMs, several metabolites had significantly increased abundances, including acetovanillone (apocynin), 12-oxo-10E-dodecenoic acid (traumatin), eriodictyol, tri hydroxycinnamoyl quinic acid, baicalin, cimicifugamide, and nicotinamide adenine dinucleotide phosphate (Appendix A). These metabolites reportedly function as antioxidants in plants [22,23,24,25,26], implying they are important for the allantoin-mediated salt stress tolerance of rice.

### 3.6. Analysis of the Correlation between the Transcriptome and Metabolome Data

To investigate the genes encoding proteins directly involved in the metabolic pathways contributing to the allantoin-mediated salt stress response of rice, we performed a KEGG pathway enrichment analysis of the DEGs and DAMs identified by the comparison between the seedlings treated with both salt and allantoin and the seedlings treated with salt alone. Four significantly enriched KEGG metabolic pathways were detected, namely ascorbate and aldarate metabolism (ko00053), cyanoamino acid metabolism (ko00460), glycerophospholipid metabolism (ko00564), and sphingolipid metabolism (ko00600) (Figure 5A). The gene and metabolite networks constructed for these pathways indicated that three genes encoding OsBGal30, HOTHEAD protein (Os06g0656000), and serine hydroxymethyltransferase (Os11g0455800) were positively correlated with serine, which is involved in cyanoamino acid metabolism (Figure 5B). Additionally, five genes (*OsBGal1*, *OsBGal3*, *Os01g0166700*, *OsBGal7*, and *OsIDD11*) correlated with serine were related to sphingolipid metabolism. Two genes encoding OsPLDα4 and phosphoesterase (Os01g0102000) were correlated with choline alfoscerate, which is involved in glycerophospholipid metabolism. Four genes (*OsALDH3E2*, *OsALDH2B5*, *OsAAO1*, and *OsAPX6*) were correlated with dehydroascorbic acid, which affects ascorbate and aldarate metabolism. These results indicate these metabolic pathways play important roles in the mechanism underlying the allantoin-mediated salt stress tolerance of rice.

## 4. Discussion

Salt is one of the major abiotic stresses limiting crop growth and productivity worldwide. Rice is sensitive to salt stress at both the seedling and reproductive stages [27]. Accordingly, improving the salt tolerance of rice plants to maintain rice production requires the thorough characterization of the diverse biological processes underlying salt tolerance. Recent research has clarified salt stress signaling pathways, including those related to the activation of stress-responsive genes, ion homeostasis, and growth regulation, as well as other downstream signal-induced responses [28]. Several studies demonstrated that the application of nitrogen or nitrogen-containing compounds could enhance salt tolerance by upregulating the ROS-scavenging system, maintaining the ion homeostasis, and increasing osmotic adjusting ability in cotton and sorghum [29,30,31]. Allantoin, an intermediate product of purine catabolism, is a common compound in plants, wherein it serves as an important nitrogen form involved in source-to-sink transport [1]. A previous study revealed that allantoin can protect plants from the detrimental effects of stress [32], but the underlying molecular mechanisms were not elucidated.

In the current study, exogenous allantoin was applied to salt-stressed rice seedlings to investigate its physiological and molecular roles during the salt stress response. Our results showed that allantoin can significantly mitigate the harmful effects of stress on rice, leading to an increase in the survival rate of rice seedlings exposed to salt stress (Figure 1B). The addition of exogenous allantoin increased the proline content, but decreased the MDA concentration, in rice seedlings under salt stress conditions (Figure 1C,D). The accumulation of proline and a decrease in the MDA content are indicators of enhanced osmotic regulation and membrane stability, both of which contribute to salt tolerance [33,34]. Furthermore, the exogenous allantoin treatment significantly increased the ROS scavenging capacity of salt-stressed rice seedlings by enhancing SOD and POD activities. This finding is consistent with the results of earlier studies, in which allantoin substantially alleviated the oxidative damage in sugar beet [13] and tomato [14]. Additionally, exogenous allantoin enhanced the root uptake of potassium and sodium but had the opposite effect on the accumulation of sodium in the shoots of rice seedlings under saline conditions. Thus, allantoin may influence the allocation and homeostasis of sodium and potassium to increase salt tolerance [35]. These findings suggest that allantoin may regulate salt stress tolerance by maintaining the osmotic equilibrium, ROS homeostasis, and sodium/potassium balance in rice, showing allantoin has the same effect on salt tolerance as nitrogen as revealed by previous studies [29,30,31].

The application of exogenous allantoin increased the endogenous allantoin level, especially in the roots of rice seedlings under salt stress conditions (Figure 1J), indicative of a positive role in the salt stress response. Allantoin is an oxidative stress marker that functions as an antioxidant, similar to vitamins [36]. However, a previous investigation concluded that allantoin does not serve as an antioxidant, even though it might be important for plant defenses against abiotic stresses [5]. In the current study, we performed a series of experiments to evaluate the antioxidant activity of allantoin in vitro. The results showed that allantoin can scavenge DPPH and ABTS free radicals, albeit not as well as vitamin C (Figure 2), implying that allantoin may have a dual role (i.e., nitrogen recycling and antioxidative activities) affecting plant growth and stress responses as previously suggested [8,37]. Exogenous allantoin may be absorbed by rice seedlings, after which it functions as an ROS scavenger, while also inducing endogenous SOD and POD to protect rice seedlings from salt stress.

Our comparative analysis of gene expression in rice seedlings treated with both salt and allantoin and seedlings treated with salt alone indicated that the applied allantoin mainly suppressed global gene expression (139 upregulated and 776 downregulated) in rice seedlings under saline conditions (Appendix A). A GO enrichment analysis revealed that a set of genes related to transport and oxidoreductase activity had upregulated expression levels following the salt + allantoin treatment (relative to the effects of salt stress alone). The genes with highly upregulated expression levels included *OsALMT4*, which affects malate transport and mineral nutrition [38], *OsCBSCLC2*, which is associated with the transport of chloride ions [39], *OsNAAT1*, which is responsible for Fe ion acquisition and transport [40], and *OsHAK4*, which helps mediate potassium transport in rice [41]. Furthermore, the allantoin-induced genes, including *OsALDH2C1*, *OsDWARF4*, *OsTRXh1*, *APX6*, and *OsHSL6* [42,43,44,45,46], were previously determined to encode proteins that regulate ROS in rice. These observations suggest that exogenous allantoin directly enhances ion transport and redox homeostasis in rice seedlings exposed to salt stress.

The comparative transcriptome analysis revealed that the salt stress and exogenous allantoin treatments resulted in genome-wide gene expression changes, whereas the salt + allantoin treatment had a smaller effect on gene expression in rice seedlings (Figure 3A). Hence, exogenous allantoin may weaken the effect of salt on gene expression in rice plants. Salt stress has adverse effects on crucial plant processes. More specifically, it can disrupt ionic equilibrium, inhibit protein synthesis and carbohydrate metabolism, and suppress plant growth and development by activating a cascade of genetic modulations [47,48]. In the present study, we detected many salt-induced genes (*n* = 608) with downregulated expression in salt-stressed rice seedlings treated with exogenous allantoin (Figure 3D). These allantoin-suppressed genes included *OsRLCK*, *OsPK*, and *OsPTK* genes, which are involved in post-transcriptional protein modifications [49,50,51]. In addition, a subset of genes encoding glycosyl hydrolase family proteins, which are mainly involved in plant carbohydrate metabolism [52], had downregulated expression levels after the exogenous allantoin treatment. The transcriptional regulation of transcription factor genes is a crucial step during plant responses to abiotic stresses [53]. In the current study, the expression of many transcription factor genes was downregulated by exogenous allantoin in salt-stressed seedlings, including *OsERF1*, *OsERF142*, *OsARF17*, *OsbZIP46*, and *OsYAB1*, which affect rice growth and development [54,55,56,57,58]. The expression levels of two classes of genes involved in chromosome remodeling (*H2A*, *H3*, and *H4*) and cell wall organization or biogenesis (*OsCESA* and *OsEXP* genes) were also clearly downregulated in response to exogenous allantoin [59,60,61]. Therefore, salt-induced changes to the expression of genes related to post-transcriptional regulation, carbohydrate metabolism, growth and development, the cell cycle, and cell wall organization might have an adverse effect on salt tolerance. The application of exogenous allantoin may minimize the effects of salt on the expression of these genes, thereby enhancing the salt tolerance of rice seedlings.

Metabolite profiling identified many metabolites that were differentially abundant between the control C18 seedlings and the seedlings treated with allantoin. The detected DAMs were associated with diverse biological pathways, implying that exogenous allantoin may affect multiple biological processes in rice seedlings under normal growth conditions, which is inconsistent with the results of an earlier study on sugar beet [13]. We observed that the effect of exogenous allantoin on the metabolites in C18 seedlings was weaker under salt stress conditions than under control conditions. Specifically, 27 metabolites accumulated more in the C18 seedlings treated with both salt and allantoin than in the seedlings treated with salt alone (Appendix A). These metabolites with increased abundances included acetovanillone, which is an important phenolic compound that regulates the redox potential during plant responses to abiotic stresses [62]. Additionally, 12-oxo-10E-dodecenoic acid (traumatin), which is a product of the oxidation of polyunsaturated fatty acids in plant tissues, promotes SOD activities in green algae under salt stress [63]. Moreover, eriodictyol, which was another metabolite that accumulated in response to exogenous allantoin, is a flavonoid with anti-inflammatory and antioxidant activities [24]. Notably, two allantoin-induced metabolites (trihydroxycinnamoylquinic acid and baicalin) were previously identified as important natural antioxidants with radical scavenging activities [45,64]. These results suggest that exogenous allantoin enhances multiple ROS scavenging pathways in rice seedlings exposed to high salinity. The exogenous allantoin treatment substantially increased the contents of inosine 5′-monophosphate and adenosine 5′-monophosphate, both of which are intermediate products of the allantoin biosynthetic pathway [32,65], implying that exogenous allantoin positively regulates the biosynthesis of endogenous allantoin in salt-stressed C18 seedlings, ultimately leading to increased salt tolerance. Thus, an increase in the allantoin content is essential for rice salt tolerance.

The combined analysis of the DEGs and DAMs revealed that a few gene-metabolite networks comprising eight genes related to serine, four genes associated with dehydroascorbic acid, and two genes related to choline alfoscerate were significantly enriched in the C18 seedlings treated with both salt and allantoin (Figure 5B–E). Serine is essential for the cell proliferation necessary for plant growth and development [66]. Dehydroascorbic acid participates in the regulation of ROS homeostasis during plant responses to abiotic stresses [67], whereas choline alfoscerate is a potential biomarker of abiotic stress responses [68] while also serving as an important nutrient source [69]. These enriched gene-metabolite networks may have crucial functions that contribute to the allantoin-mediated salt stress tolerance of rice.

In conclusion, exogenous allantoin positively affects the rice salt tolerance at the seedling stage by increasing the contents of endogenous allantoin, increasing the reactive oxygen species (ROS) scavenging capacity, and maintaining sodium and potassium homeostasis while also mediating the global reprogramming of the transcriptome (such as ion transport, redox regulation, and many salt-induced genes) and modulating downstream metabolic pathways. These results have clarified the transcript-level and metabolic processes underlying the allantoin-mediated salt tolerance of rice.

## Figures and Tables

**Figure 1 antioxidants-11-02045-f001:**
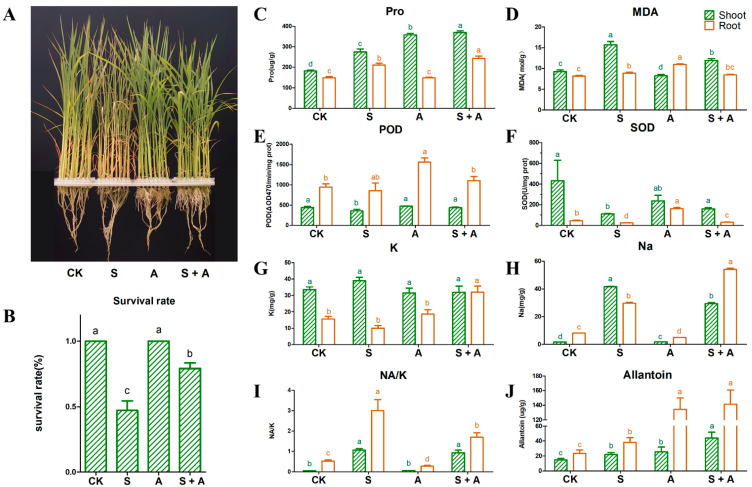
Phenotypic and physiological analyses of C18 seedlings under different treatment conditions. Phenotypes of the C18 seedlings in response to different treatments (**A**). Survival rates of C18 seedlings after a 14-day treatment and a 7-day recovery period (**B**). Proline (**C**), MDA (**D**), POD (**E**), SOD (**F**), K^+^ (**G**), and Na^+^ (**H**) contents and Na^+^/K^+^ ratio (**I**) in the shoots and roots after different treatments. Allantoin contents in the shoots and roots after different treatments (**J**). The rice seedlings were maintained under control conditions (CK) or were treated with 140 mM NaCl (S), 10 mM allantoin (A), or 140 mM NaCl + 10 mM allantoin (S + A) for 7 days. Letters a,b,c, and d represent the results of significant difference analysis. The same letter marks indicate no significant difference between two groups of data, while different letters indicate a significant difference between two groups of data (*p* < 0.05).

**Figure 2 antioxidants-11-02045-f002:**
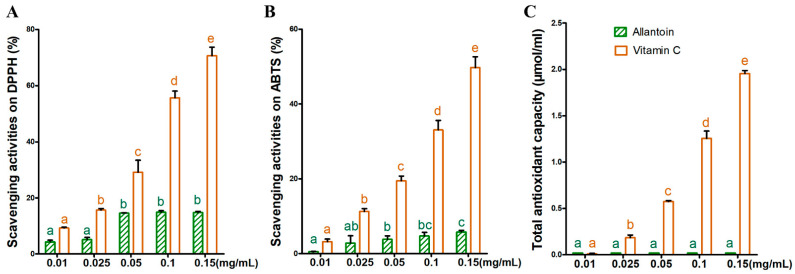
Determination of the antioxidant activity of allantoin in vitro. Evaluation of the ability of allantoin and ascorbic acid to scavenge DPPH free radicals (**A**) and ABTS free radicals (**B**) as well as their total antioxidant capacity (T-AOC) (**C**). Letters a,b,c,d and e represent the results of significant difference analysis. The same letter marks indicate no significant difference between two groups of data, while different letters indicate a significant difference between two groups of data (*p* < 0.05).

**Figure 3 antioxidants-11-02045-f003:**
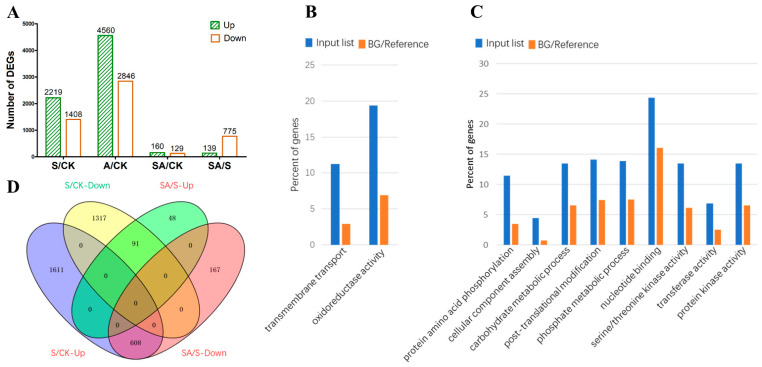
Overview of the transcriptome profiling analysis and the GO enrichment analysis of the differentially expressed genes (DEGs) in C18 seedlings. The DEGs in the C18 seedlings revealed by the salt (S) vs. control (CK), allantoin (A) vs. CK, SA vs. CK, and SA vs. S comparisons (**A**). The enriched GO terms among the upregulated genes (**B**) and the downregulated genes (**C**) revealed by the SA vs. S comparison. A Venn diagram was constructed to analyze the up- and downregulated genes detected by the S vs. CK and SA vs. S comparisons (**D**).

**Figure 4 antioxidants-11-02045-f004:**
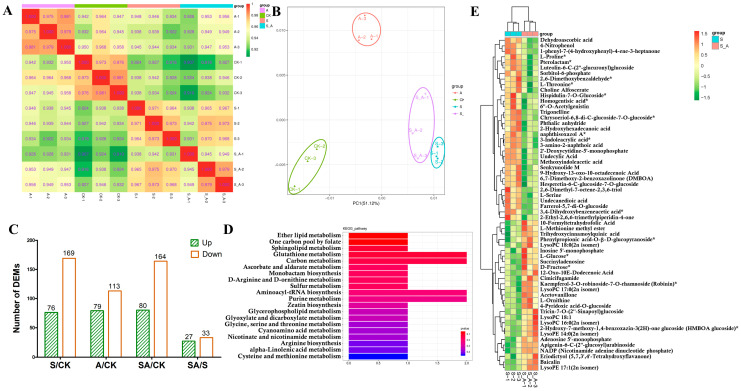
Metabolite profiling analysis of the C18 seedlings under salt (S), allantoin (A), salt + allantoin (SA), and control (CK) conditions. Correlation analysis of the biological samples for the four treatments (**A**). Principal component analysis of the metabolome data for the four treatments (**B**). Overview of the differentially abundant metabolites revealed by the S vs. CK, A vs. CK, SA vs. CK, and SA vs. S comparisons (**C**). Results of the KEGG enrichment analysis and a cluster heat map of the differentially abundant metabolites in the C18 seedlings revealed by the SA vs. S comparison (**D**,**E**). The * in E indicates that this metabolite is an isomer.

**Figure 5 antioxidants-11-02045-f005:**
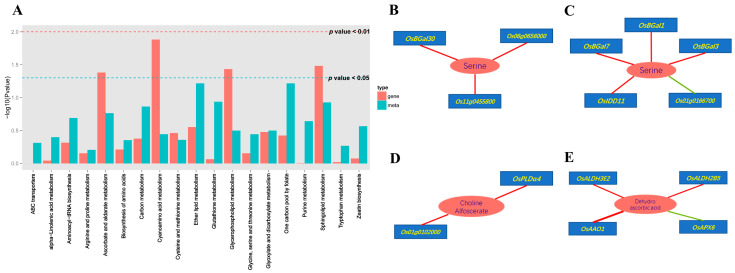
Correlations between the transcriptome and metabolome data. Results of a KEGG co-enrichment analysis of the differentially expressed genes and differentially abundant metabolites in the C18 seedlings revealed by the comparison between the salt + allantoin treatment and the salt stress treatment (**A**). Green and red lines indicate the selected genes and metabolic pathways at *p* < 0.05 and *p* < 0.01, respectively. Cyanoamino acid metabolism (**B**). Sphingolipid metabolism (**C**). Glycerophospholipid metabolism (**D**). Ascorbate and aldarate metabolism (**E**). The circles represent metabolites, the boxes represent genes, and the lines represent correlations (positive and negative correlations are in red and green, respectively).

## Data Availability

Not applicable.

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
