# Peer review of "Transcriptome and Metabolome Analyses Reveal Complex Molecular Mechanisms Involved in the Salt Tolerance of Rice Induced by Exogenous Allantoin"

_antioxidants, 2022, doi:10.3390/antiox11102045_

Round 1

Reviewer 1 Report

The manuscript by Juan Wang et al "Transcriptome and metabolome analyses reveal complex mo- 2 lecular mechanisms involved in the salt tolerance of rice in- 3 duced by exogenous allantoin" is an interesting piece of work that links nitrogen recycling/metabolism in plants to salt stress toleranxce.

The presentation of the experimental results is clear and well organized. The authors need to specify the number of plants they used for the survival analysis.

Author Response

1: The presentation of the experimental results is clear and well organized. The authors need to specify the number of plants they used for the survival analysis.

Thanks for the suggestions and comments. We have added the detail information for the plants used in this study in the Materials and Methods.

" 144 rice seedling plants were used for each treatment. Each treatment was replicated three times."

Reviewer 2 Report

      This study investigate the effects of exogenous allantoin application on physiological indices, transcriptomes, and metabolomes in rice seedlings under salt stress. The authors show the mechanisms of the alleviative effects of allantoin by enhancing ROS scavenging cascades and up-regulating transmembrane transport, leading to less damage and lower Na contents in cells, respectively. This report is suitable for the publication in Antioxidants. Please consider my suggestion for a revision as described below.

L52-53
      This article [16, Selamoglu et al.], which is cited by the authors, shows the in-vitro antioxidant activities of some contained-allantoin plants using the plant-extracts. Thus, it is understandable for readers to describe more detail. Please rephrase.

L77
      The more detailed description about the cultivar of C18 such as the degree of salt stress tolerance is needed.

L82
      The rice seedlings were grown and treated with salt stress in a greenhouse condition. However, more detailed descriptions such as temperature, light intensity, and the length of photoperiod are needed. The effects of salt stress may change by the environmental conditions.

L84
      The rice seedlings were treated with 140 mM NaCl for several days in this study. Why did the authors decide the concentration of salt stress? I think that the concentration is high for rice. If the authors conducted preliminary tests, the data should be shown as supplementary. Or, if the authors published the data of C18 under salt stress, please add the citation.

L93
      The author describe “After the plants were treated for 14 days, they were transferred to Yoshida nutrient solution for a 7-day recovery period”. In the legend in Figure 1, however, there is the description of “C18 seedlings after a 7-day treatment and a 7-day recovery period”. Which descriptions are correct?

Discussion
      It has been well investigated that the application of nitrogen is effective to ameliorate the adverse effects of salt stress on plants (Mansour, 2000, Biol Plant, 43: 491-500; Esmaili et al., 2005, Plant Soil Environ, 54: 537-546). Recently, nitrogen supplementation is effective to alleviate the salt-induced oxidative damage by up-regulating antioxidant defense system (Sikder et al., 2020, Plants, 9(4): 450, doi: 10.3390/plants9040450). This result may be similar to the application with allantoin. The author should discuss the similarity and difference between the application with nitrogen and the nitrogenous compound of allantoin in the Discussion.  

Figures 3, 4, and 5
      These results are important for this manuscript. However, as the characters are too small and the resolution is too low, it is difficult to understand. The supplemental figures 2 and 3 are understandable, and thus I think that the authors can improve the resolution of characters.
